# An Artificial Plant Community Algorithm for the Accurate Range-Free Positioning of Wireless Sensor Networks

**DOI:** 10.3390/s23052804

**Published:** 2023-03-03

**Authors:** Zhengying Cai, Shan Jiang, Jiahuizi Dong, Sijia Tang

**Affiliations:** College of Computer and Information Technology, China Three Gorges University, Yichang 443002, China

**Keywords:** wireless sensor networks, positioning algorithm, Internet of Things, artificial intelligence, artificial plant community

## Abstract

The problem of positioning wireless sensor networks is an important and challenging topic in all walks of life. Inspired by the evolution behavior of natural plant communities and traditional positioning algorithms, a novel positioning algorithm based on the behavior of artificial plant communities is designed and presented here. First, a mathematical model of the artificial plant community is established. Artificial plant communities survive in habitable places rich in water and nutrients, offering the best feasible solution to the problem of positioning a wireless sensor network; otherwise, they leave the non-habitable area, abandoning the feasible solution with poor fitness. Second, an artificial plant community algorithm is presented to solve the positioning problems encountered in a wireless sensor network. The artificial plant community algorithm includes three basic operations, namely seeding, growing, and fruiting. Unlike traditional artificial intelligence algorithms, which always have a fixed population size and only one fitness comparison per iteration, the artificial plant community algorithm has a variable population size and three fitness comparisons per iteration. After seeding by an original population size, the population size decreases during growth, as only the individuals with high fitness can survive, while the individuals with low fitness die. In fruiting, the population size recovers, and the individuals with higher fitness can learn from each other and produce more fruits. The optimal solution in each iterative computing process can be preserved as a parthenogenesis fruit for the next seeding operation. When seeding again, the fruits with high fitness can survive and be seeded, while the fruits with low fitness die, and a small number of new seeds are generated through random seeding. Through the continuous cycle of these three basic operations, the artificial plant community can use a fitness function to obtain accurate solutions to positioning problems in limited time. Third, experiments are conducted using different random networks, and the results verify that the proposed positioning algorithms can obtain good positioning accuracy with a small amount of computation, which is suitable for wireless sensor nodes with limited computing resources. Finally, the full text is summarized, and the technical deficiencies and future research directions are presented.

## 1. Introduction

Modern industry and society make extensive use of various Internets of Things (IoTs) and wireless sensor networks (WSNs) [1], where many small wirelessly interconnected devices and sensor nodes are equipped with a microprocessor, transceiver, and power unit. Hence, user applications can send data to a base station for information gathering and analysis; position information is one of the most important aspects of these datasets [2]. Nowadays, position-based applications are popular in various industries, including industry 4.0 [2], smart grids [3], geographic routing [4], traffic management [5], unmanned underwater vehicles (UUVs) [6], and so on. Applications based on geographical positioning can enhance user personalized services to a new level, improve network security [7], prevent many network crimes to a large extent, help digital marketers to improve business conversion rates, and achieve search engine optimization.

Many industrial applications need to associate detected events with their precise geographical positions, which creates higher requirements for accuracy, real-time analysis, and costs [1,8]. In WSNs, according to whether the nodes need to measure the distance or angle between nodes during the positioning process, the positioning algorithm can be divided into two types: range-based algorithms and range-free algorithms.

Range-based positioning algorithms need to obtain information about the distance between nodes and other related information with the help of additional hardware during localization, and then select the appropriate calculation method to calculate the coordinates of the nodes to be solved. Range-based algorithms include Time of Arrival (ToA) [9,10,11,12], Time Difference of Arrival (TDoA) [7,12,13], Received Signal Strength Intensity (RSSI) [7,11,14] and Angle of Arrival (AoA) [9,10], among others. This kind of range-based method has good positioning accuracy and efficiency, but also entails high costs and energy consumption.

The range-free positioning algorithm does not require additional hardware, and it selects the appropriate calculation method for positioning the unknown nodes according to the node information communication situation. Moreover, most applications prioritize the requirements of energy consumption, hardware, and the cost of sensor nodes, hoping to meet the needs of the application with limited energy and simple hardware. For most wireless sensor network applications, the accuracy of node location does not need to be very high and range-free positioning algorithms are popular. Range-free positioning algorithms include the Distance Vector (DV-Hop) algorithm [2,15], the Centroid algorithm [14], the Approximate Point in Triangle Test (APIT) algorithm [16], and the Amorphous algorithm [17].

To improve the positioning accuracy and solving performance of these positioning algorithms, many artificial intelligence (AI) and heuristic algorithms are widely used; these include Artificial Neural Networks (ANN) [6], Convolutional Neural Networks (CNN) [18,19], Deep Learning (DL) [20,21], Fuzzy Logic (FL) [22], Particle Swarm Optimization (PSO) [4], Ant Colony Optimization (ACO) [22], the Artificial Bee Colony (ABC) [8], the Genetic Algorithm (GA) [23], the Artificial Fish Swarm Algorithm (AFSA) [24], and Simulated Annealing (SA) [25]. These hybrid AI and range-free algorithms show great advantages in wireless sensor network positioning. 

The application of artificial intelligence in range-free algorithms has recently become a hot research topic. Therefore, our research team was drawn to the field and attempted to develop an effective method for solving the positioning problem of WSNs. Most artificial intelligence algorithms focus on the learning mechanisms of nervous organisms, i.e., ANN [6], CNN [18,19], DL [20,21], FL [22], PSO [4], ACO [22], ABC [8], GA [23] and AFSA [24], but few studies have considered the learning behaviors of plants. Plant communities have survived on the earth for hundreds of millions of years, and they must also have their own evolutionary strategies. Regardless of the fact that plant individuals have no nerves, they can also search for habitable areas rich in both water and nutrients through the continuous cycle of seeding, growing, and fruiting [26]. Therefore, we attempted to simulate the evolution behavior of plant communities on a personal computer; the proposed artificial plant community (APC) algorithm has the potential to solve the positioning problem of WSNs. Unlike traditional artificial intelligence algorithms, which always have a fixed population size and only one fitness comparison per iteration, the artificial plant community algorithm has a variable population size and three fitness comparisons per iteration. Our objective was to tap the potential of plant evolution mechanisms; this is very simple in theory, but it can better preserve the optimal solution in each step of the calculation, and it also retains a certain global search capability.

The main contributions of this paper are as follows.

First, the positioning problem of WSNs is established as an evolution model of an artificial plant community. The habitable area, which is rich in water and nutrients, is described as a fitness function for the artificial plant community’s searching and evolution. Hence, the anchor nodes are detected and the unknown nodes are positioned.

Second, an artificial plant community algorithm is illustrated to simulate the growth process of a true plant community and to solve the fitness function. The main behaviors of all individuals in an artificial plant community comprise three operations: seeding, growing, and fruiting. When seeding, the individuals or seeds are generated according to the predetermined population size. During growth, the population size changes a little. Only the individuals with high fitness can survive, while the individuals with low fitness die. During fruiting, the population size recovers, and the individuals with higher fitness can produce more fruits. The optimal solution in each iterative computing process can be preserved as a parthenogenesis fruit for the next seeding operation. When seeding again, the fruits with high fitness can survive and be seeded, while the fruits with low fitness die, and a small number of new seeds are generated through random seeding. After iterative computing of the three operations, the artificial plant community obtains the accurate positions of the unknown nodes.

Third, the proposed APC algorithm is verified using a series of experiments and is compared with other AI positioning algorithms. Comparative analysis results are provided and discussed.

The rest of the paper is organized as follows. Section 2 presents the background and the related literature on positioning algorithms. In Section 3, the artificial plant community is introduced. In Section 4, the positioning process of the APC algorithm is illustrated and the positioning fitness function is built. In Section 5, the experimental results are analyzed and performance comparisons are discussed. Finally, Section 6 offers some conclusions and discusses the limitations and future directions of this research.

## 2. Background and Related Works

The nodes in wireless sensor networks can be divided into two main types: anchor nodes or reference nodes, and unknown nodes [6,27]. The anchor node has a well-known position, which can help to determine the positions of unknown nodes, and the positions of the other unknown nodes can be determined only through anchor nodes. 

The Global Positioning System (GPS) is undoubtedly the most accurate method for estimating the positions of sensor nodes; it functions by equipping a GPS device [28]. However, it is often not the best choice for WSNs because of the high cost of additional GPS devices and very high requirements for GPS signals. GPS signals are not available in many cases, such as in GPS-denied areas [7]. Due to the high power consumption of GPS devices, the batteries in the wireless sensors, which have limited capacity, are quickly exhausted. 

Range-based algorithms include Time of Arrival (ToA) [9,10,11,12], Time Difference of Arrival (TDoA) [7,12,13], Received Signal Strength Intensity (RSSI) [7,11,14,29] and Angle of Arrival (AoA) [9,10]. All of these range-based positioning algorithms can compute the distance or the angle between synchronized sensor nodes, and then estimate the position of unknown nodes using trilateration or triangulation methods. These methods often require extra hardware devices and the high time synchronization of each anchor node, and the power consumption of the sensor nodes is often related to the calculation, transmission, and storage of the positioning algorithm.

Range-free algorithms use the anchor nodes with assigned exact geographic positions to calculate the positions of unknown nodes, but rely only on the connectivity information between sensor nodes [30,31,32]. As such, range-free algorithms first need to search for connectivity with nearby anchors in the same geographic topology and then speculate the positions of unknown nodes in the wireless sensor network. Range-free positioning algorithms include the Distance Vector (DV-Hop) algorithm, the Centroid algorithm, the Approximate Point in Triangle Test (APIT) algorithm, and Amorphous. 

DV-Hop [2,15,27,30] can compute the minimum number of hops towards the anchors to the unknown sensor nodes and calculate the positions of unknown nodes by the average hop size. This algorithm is easy to implement without requiring additional overheads, and it offers high-quality coverage. DV-Hop is a popular range-free positioning algorithm that is suitable for the positioning of unknown WSN nodes with some neighbor anchors. Additionally, DV-Hop has the simple topology requirement that unknown nodes are surrounded by at least three neighbor anchors.

The Centroid algorithm [14] estimates the positions of unknown sensor nodes, without additional materials, as the barycenter surrounded by a number of neighboring anchor nodes. The authors of [14] presented an RSSI-based hybrid centroid-k-nearest-neighbors localization method. The positioning accuracy of the Centroid algorithm is not high and is greatly affected by the density of tracing nodes. Only in high-density networks can the positioning effect of the centroid algorithm be reflected.

The APIT algorithm [16,32] computes three anchor nodes to estimate the positions of unknown nodes and can achieve better performance in the case of a great number of anchors. The authors of [32] introduced a virtual nodes-based range-free APIT localization scheme for WSN. In APIT positioning, unknown nodes need to communicate with neighboring nodes, which requires a high node density and anchor node density and is vulnerable to noise and other factors. Its positioning accuracy mainly depends on the number of triangles containing unknown nodes. In a network with a high number of anchor nodes, the positioning effect may be good.

The Amorphous algorithm [17] is similar to the DV-Hop algorithm and easily attains high accuracy when estimating the positions of unknown nodes, especially in random WSNs. However, it requires a high density of anchor nodes, and the value of network connectivity must be obtained before positioning, which is not conducive to the expansion of the algorithm.

On the whole, range-free algorithms have several advantages over GPS methods and range-based algorithms, including low costs, low communication overheads, the lack of additional requirements for equipment, low power consumption, and better positioning accuracy. Recently, many scholars have used artificial intelligence methods to improve the accuracy of range-free algorithms. The authors of [6] proposed reinforcement learning (RL) and neural network to optimize the selection of anchor nodes for mobile underwater positioning. Meanwhile, [31] calculated RSS information between the target nodes and the corresponding anchor nodes to determine the positions of target nodes based on edge weights, where the edge weights are adjusted by hybrid Grey Wolf Optimization with a Firefly Algorithm (GWO-FA). In these AI algorithms, each individual in a swarm has only simple intelligence, but they can exhibit complex swarm intelligence through learning and cooperation with each other [33].

Therefore, in recent years, the application of artificial intelligence and swarm intelligence methods to improve the performance of range-free algorithms has attracted increasing amounts of attention, and this field has become a hot research topic. Related applications of artificial intelligence algorithms include Artificial Neural Networks (ANN) [6,29], Convolutional Neural Networks (CNN) [18,19], Deep Learning (DL) [20,21], Fuzzy Logic (FL) [22,34], Particle Swarm Optimization (PSO) [4,35,36], Ant Colony Optimization (ACO) [22,37], Artificial Bee Colony (ABC) [8,29], the Genetic Algorithm (GA) [23,38], the Artificial Fish Swarm Algorithm (AFSA) [24], Simulated Annealing (SA) [25], marine predator optimization [4], the krill herd optimization algorithm [17], Grey Wolf Optimization (GWO) [31], the Firefly Algorithm (FA) [31], the aquila optimization algorithm [39], the artificial hummingbird algorithm [40], the Phasmatodea population evolution algorithm [41], the Sparrow Search Algorithm (SSA) [42], the Coot Bird Algorithm (CBA) [43], the Fruit Fly Optimization Algorithm (FOA) [44], the Whale Optimization Algorithm (WOA) [45], and the Jaya algorithm [46]. In these positioning algorithms, a swarm comprises a number of simple individuals who achieve a search function and complete the positioning task together, through simple cooperation with each other.

As the above summary indicates, this is a hot research topic; applying AI to WSN positioning has great potential to optimize the solving performance and minimize errors, especially in the area of range-free algorithms. The authors of [47] undertook a systematic review of localization in WSN from the perspectives of machine learning and optimization-based approaches; they indicated that many kinds of optimization methods have great prospects, since wireless sensors’ network locations are affected by many factors, including energy efficiency, accuracy, errors, and complexity. However, a few researchers have noted that nerveless plant communities also possess swarm intelligence, allowing them to effectively solve all kinds of complex problems, as demonstrated by the artificial slime mold algorithm [5] and the artificial Physarum swarm algorithm [48]. A plant community without nerves can also use the swarm learning mechanism and evolutionary computing to adapt to the environment [49]. In the process of survival, natural plant communities prosper or decline when subjected to different external environments. This characteristic of variable population size is not accounted for by traditional artificial intelligence algorithms, which can help us to quickly screen out the optimal solution and retain sufficient global search abilities. Therefore, a novel positioning algorithm based on DV-Hop and an artificial plant community (APC) is proposed, with the aim of increasing the positioning accuracy of unknown nodes in WSNs.

## 3. Proposed Positioning Algorithms

In this section, we introduce a new positioning algorithm based on the DV-Hop and artificial plant community algorithm. The DV-Hop positioning algorithm [2,15,27,30] can compute the average hop size to undertake position estimation for most WSN applications. Hence, the proposed APC is a kind of distributed algorithm and is suitable for efficiently improving the position accuracy of the DV-Hop algorithm. Here, we first describe the basic structure and main operations of APC, and then the positioning algorithm based on APC is illustrated.

### 3.1. Basic Structure of APC

According to the natural plant community [26,49], the basic structure of an artificial plant community system includes many artificial plant individuals as feasible solutions and the living area as a whole solution space, as shown in Figure 1. Furthermore, the living area can be divided into habitable zones and non-habitable zones. In the evolutionary process, the plant community can only survive in habitable zones that are rich in water and nutrients, namely the spaces of feasible solutions. The other, non-habitable zones are not suitable for the natural plant community to live in; these zones constitute the space of infeasible solutions. An artificial plant community requires a fitness function to distinguish the habitable zones and non-habitable zones.

An artificial plant community is composed of a number of artificial plant individuals, where each artificial plant individual has the same characteristics and behaviors. An artificial plant individual may be a seed in seeding operation or a fruit in fruiting operation. There is no central control in the artificial plant community, and the individual computing is distributed. As such, it is highly suitable as a working environment for wireless sensor networks and has strong scalability, where the failure of one or several individuals does not affect the solving process of the whole artificial plant community. Each artificial plant individual can search and learn the environment, and they can indirectly communicate with each other. Therefore, when the number of artificial plant individuals expands, the rise in communication costs is small. The APC algorithm has good scalability in positioning problem solving, where the artificial plant community evolves and learns information through indirect communication.

The living space includes habitable zones and non-habitable zones, which are identified by solving problems. Each artificial plant individual is randomly distributed across the whole living area and then marks a habitable zone or a non-habitable zone according to its learning experience. The swarm learning ability or evolution behavior of each artificial plant individual is so simple that the artificial plant community algorithm can easily be implemented. The learning behavior of the APC is a kind of emergent intelligence generated through the interaction processes of different individuals. A fitness function acts as a medium for interactive information, and each artificial plant individual can learn environmental information by comparing fitness function values. After many iterations, the artificial plant community can detect habitable zones and non-habitable zones to estimate the positions of known sensor nodes.

In the proposed positioning model, all the feasible solutions are based on a fitness function of DV-hop. The artificial plant individuals are randomly distributed throughout the whole wireless sensor network with anchor nodes and unknown nodes. Each artificial plant individually searches the distance vectors between the anchors and unknown nodes according to their swarm learning and evolutionary mechanisms. After several iterative calculations, the artificial plant community can determine the optimal solution as a feasible position for an unknown node.

### 3.2. Symbol Definitions

In this section, the symbol definitions are provided that are used in the next sections, as shown in Table 1.

For a sensor positioning problem, the sensor node set is *N* = {*N_i_*|*i* = 1,2,…,*n*}, where the anchor node set is *A* = {*A_i_*|*i* = 1,2,…,*a*} and the unknown node set is *U* = {*U_i_*|*i* = 1,2,…,*n*-*a*}. The value of the hop counter on the hello packet from the anchor node *A_i_* to the normal sensor node *Nj* is *h_ij_*. The value of *h_ij_* is continuously updated throughout the whole positioning computation until it records the minimum number *h_ij_*_min_ of hops between the anchor *A_i_* and the normal node *N_j_*. The average value of the hop counter on nodes *N_i_* is calculated as *avgh_i_*, and the estimated distance between nodes *N_i_* and *N_j_* is *d_i__j_*. The coordinates (*x_i_*, *y_i_*) of node *N_i_* are the ultimate goal of the positioning problem in WSNs.

For geographic location information, we can use the real-number-encoding method for variable *x*. The iteration counter *k* is used to count the number of iterative computations with the maximum value *K*. In each iterative computing, the APC uses a function of *fitness* to learn and evolve. The seeding probability *p_s_*, growing probability *p_g_*, and fruiting probability *p_f_* help to instruct the APC to search for the optimal solution. If the positioning error *e* reaches the error threshold *e_th_*, then the iterative computing is stopped and the optimal solution is output.

### 3.3. Assumptions

In this section, some assumptions are made to simplify the algorithm design.

i.It is assumed that the category difference of artificial plant individuals is not considered in the artificial plant community. Different artificial plant individuals have the same characteristics and operations of seeding, growing, and fruiting, and there are no barriers to information exchange between them. Competition between plant community species is not considered here.ii.It is assumed that the influence of light is not considered and that the artificial plant community only requires water and nutrients. Areas with water and nutrients are habitable zones for an artificial plant community and the light difference between day and night is assumed not to affect the survival of an artificial plant community.iii.It is assumed that the population size of an artificial plant community is controlled in this algorithm. The natural plant community may become more prosperous or decline during its growth, and the population size may become larger or smaller. The artificial plant community also obeys the principles of stability, where the population size shrinks in the growing process and expands in the fruiting process.iv.It is assumed that the seeding operation is random. The seeding ranges of an artificial plant community are uncertain, simulating the influence of media organisms, wind, water, rain, snow, fire, etc.v.It is assumed that the growing operation is determined by the fitness function of solving problems. For the problem of sensor node positioning, the fitness function can be designed to measure the distance vectors of hop.vi.It is assumed that a pair of artificial parents will produce two new artificial children each time, and that an artificial parent can also produce a new artificial child to simulate parthenogenesis in a natural plant community. All the artificial plant individuals are assumed to be fruit according to the proximity principle by a fruiting probability.

### 3.4. Main Operations of APC

An artificial plant community has three main operations: seeding, growing, and fruiting.

The seeding operation is a random process of searching for feasible solutions across the whole solution space, where an artificial plant community may be distributed in any place by a seeding probability *p_s_*, as shown in Figure 1a. Each artificial plant individual can be called a seed in seeding operation and is randomly scattered in the solution space in the form of seeds at the beginning, but not necessarily in the habitable zones. A seed may become an artificial plant individual during the growing operation and a fruit during the fruiting operation. The seeding operation can help the artificial plant community to find more feasible solutions according to the seeding probability.

The growing operation is a process of natural selection; the artificial plant individuals not scattered in the habitable zone will die, as shown in Figure 1b. Not all artificial plant individuals or seeds can grow after random seeding. Only the artificial plant s seeds in habitable zones may grow into individuals by a growing probability *p_g_*, but the artificial plant individuals seeded in non-habitable zones are eliminated naturally. The artificial plant community algorithm judges the habitable zones using a fitness function.

The fruiting operation is a swarm learning process, and each artificial plant individual has the opportunity to learn from the experience of other individuals, as shown in Figure 1c. A fruit may become a seed in the feeding operation and an artificial plant individual in the growing operation. After seeding and growing, the surviving artificial plant individuals often have high values of the fitness function. Two parents can fruit and give birth to two children by learning from each other’s experiences. A single parent can also fruit to bear a child, which simulates the parthenogenesis of natural plants. The new individuals have many characteristics of the parents, and the part inherited from the parents depends on the fruiting probability *p_f_*.

## 4. APC-Based DV-Hop Positioning Algorithm

DV-Hop is a popular range-free positioning algorithm and is suitable for positioning unknown sensor nodes according only to some nearby anchor nodes. In this section, a novel APC-based positioning algorithm for WSNs is introduced. The algorithm includes four main stages, as follows:i.Computing the minimum hop. The anchor nodes flood their positions, and every anchor node can broadcast a packet containing its position information. After several broadcasts, the minimum hop is calculated for each sensor node.ii.Calculating the average hop size. The weighted mean technique can be used to speculate the average hop size according to the positions of the anchor nodes. Then, the average hop distance can be computed by the anchor nodes.iii.Estimating the positions of unknown nodes. The multi-trilateration technique is often used to estimate the possible positions of unknown nodes in WSNs.iv.APC optimization. The APC is used to accurately search for the optimal positions of unknown nodes. The APC algorithm can be combined with the DV-Hop method to accurately determine the positions of unknown nodes in wireless sensor networks.

### 4.1. Computing the Minimum Hop

The DV-Hop algorithm [2,15,27,30] is suitable for sensor positioning in WSNs and is often considered a benchmark in WSN applications. The DV-Hop algorithm first needs to obtain the positions of several nearby anchor nodes and then estimate the positions of the unknown nodes, where an unknown node is a sensor node with unknown coordinates to be estimated. Anchor nodes often obtain accurate coordinates by means of carrying positioning equipment, or by other means, in order to obtain accurate position coordinates in advance. There are three main stages in the DV-Hop algorithm, namely computing the minimum hop, calculating the average hop size, and estimating the positions of unknown nodes.

To compute the minimum hop, a hop counter should be first initialized to 0. The value *h_ij_* of the hop counter on the hello packet is initialized, and it records the minimum number of hops between anchor *A_i_* and every node *N_j_* (an anchor node *A_i_* or unknown node *U_i_*). Then, all the anchors *A_i_* broadcast a hello packet containing their exact position information; at the same time, the hop counter continuously increases with the increase in hops during the rebroadcasting. In the first reception of the hello packet, every node *N_j_* records the position of anchor *A_i_*. In the following receptions, if the same hello packet is received repeatedly, every node *N_j_* will decide whether to update the value of *h_ij_*. If the value of the hop counter in the newly received packet is lower than the recorded value *h_ij_*, the node *N_j_* will update the recorded value *h_ij_* with the lower value of the hop counter in the newly received packet and relieve the packet; otherwise, the node *N_j_* will neglect the packet. After the broadcast, each sensor node *N_j_* reports the minimum number *h_ij_*_min_ of hops to all anchor nodes *A*.

### 4.2. Calculating the Average Hop Size

Assuming that there is an anchor node *A_i_* with coordinates of (*x_i_*, *y_i_*) and a normal node *N_i_* with coordinates of (*x_j_*, *y_j_*), the distance *d_i__j_* between the two nodes *A_i_* and *N_j_* is computed as follows:(1)$$d_{ij}^{}=\sqrt{{\left(x_{i}-x_{j}\right)}^{2}+{\left(y_{i}-y_{j}\right)}^{2}}.$$

A normal node *N_i_* may be an anchor node or an unknown node. If it is an unknown node, the following steps can help to determine its location. If it is an anchor node, the following steps can also help it to proofread and update its location, which is particularly useful in fast-moving and dynamic sensor networks.

According to the minimum number *h_ij_*_min_ of hops and the distance *d_i__j_* between the two nodes *A_i_* and *N_j_*, the average hop size *avgh_j_* can be calculated. Statistical methods can be used to count the error in the hop data and help us to calculate the value of average hop size *Avgh_j_* on node *N_j_*. For the total number *a* of anchor nodes, the statistical error *s_j_* can be estimated.
(2)$$s_{j}=\frac{1}{a-1}\sum_{i\neq j}\left(d_{ij}-Avgh_{j}\times h_{ij}^{}\right).$$

Then, the first derivative of Equation (2) is used to determine the extreme value, and the extreme value is 0.
(3)$$\frac{\partial s_{j}}{\partial Avgh_{j}}=0.$$

Then, the average hop size *avgh_j_* on node *N_j_* can be calculated as follows:(4)$$Avgh_{j}=\frac{\sum_{j\neq i}h_{ij}\times d_{ij}}{\sum_{j\neq i}h_{ij}^{2}}$$

Here, the weight is introduced into the DV-Hop algorithm to decrease the positioning error, and the weight *w_j_* on node *N_j_* can be calculated as shown in Equation (5):(5)$$w_{j}=\frac{1}{\sum_{i\neq j}\left|Avgh_{i}-Avgh_{j}\right|}.$$

Now, the weighted average hop size $Avgh_{j}^{’}$ can be obtained as follows:(6)$$Avgh_{j}^{’}=\frac{\sum_{i=1}^{a}w_{j}\times Avgh_{j}}{\sum_{i=1}^{a}w_{j}}.$$

### 4.3. Estimating the Positions of Unknown Nodes

In the third step, a multi-trilateration positioning method is introduced to estimate the positions of unknown nodes. According to the reception of $Avgh_{j}^{’}$, the normal node *N_j_* can multiply the hop number of *h_ij_* to *A_i_*. Because the anchor node is also a kind of normal node, the approximate distance *d_i_* of each anchor node *A_i_* can be obtained.
(7)$$d_{i}=Avgh_{i}^{’}\times h_{ij}.$$

Because there is a total of *a* anchor nodes, the following equation can be used to estimate the coordinates (*x_j_*, *y_j_*) of *N_j_*:(8)$$\left\{\begin{array}{c} {\left(x_{j}-x_{1}\right)}^{2}+{\left(y_{j}-y_{1}\right)}^{2}=d_{1}^{2}\\ {\left(x_{j}-x_{2}\right)}^{2}+{\left(y_{j}-y_{2}\right)}^{2}=d_{2}^{2}\\ \vdots \\ {\left(x_{j}-x_{a}\right)}^{2}+{\left(y_{j}-y_{a}\right)}^{2}=d_{a}^{2}. \end{array}\right.$$

Based on least-squares techniques, Equation (8) can be solved and the position *d_j_* of a normal node *N_j_* can be obtained:(9)$$d_{j}=\left[\begin{array}{c} x_{j}\\ y_{j} \end{array}\right]={\left(U^{T}V\right)}^{-1}U^{T}V,$$
where
(10)$$U=-2\times {\left[\begin{array}{cc} x_{1}-x_{a} & y_{1}-y_{a}\\ x_{2}-x_{a} & y_{2}-y_{a}\\ \vdots & \vdots \\ x_{a-1}-x_{a} & y_{a-1}-y_{a} \end{array}\right]}_{,}$$
(11)$$V=\left[\begin{array}{c} d_{1}^{2}-d_{a}^{2}-x_{1}^{2}+x_{a}^{2}-y_{1}^{2}+y_{a}^{2}\\ d_{2}^{2}-d_{a}^{2}-x_{2}^{2}+x_{a}^{2}-y_{2}^{2}+y_{a}^{2}\\ \vdots \\ d_{a-1}^{2}-d_{a}^{2}-x_{a-1}^{2}+x_{a}^{2}-y_{a-1}^{2}+y_{a}^{2} \end{array}\right].$$

Here, some intermediate symbols are used to simplify expressions and do not represent physical quantities of special significance. *U^T^* represents the transpose of the matrix *U*, and *U*^−1^ denotes its inverse. The anchor nodes should not be on the same line; otherwise, the matrix operation *U^T^U* will be singular and the matrix operation (*U^T^U*)^−1^ will not exist.

If we note that $P_{i}=x_{i}^{2}+y_{i}^{2}$ and $Q_{j}=x_{j}^{2}+y_{j}^{2}$, then Equation (8) can be rewritten in a new form:(12)$$d_{ij}^{2}-P_{i}^{2}=-2x_{i}x_{a}-2y_{i}y_{a}+S_{j}.$$

According to Equation (10), we note two intermediate matrixes, as follows:(13)$$F={\left[\begin{array}{ccc} -2x_{1} & -2y_{1} & 1\\ -2x_{2} & -2y_{2} & 1\\ \vdots & \vdots & \vdots \\ -2x_{a} & -2y_{a} & 1 \end{array}\right]}_{,}$$
(14)$$G={\left[\begin{array}{c} d_{ij}^{2}-R_{1}\\ d_{ij}^{2}-R_{2}\\ \vdots \\ d_{aj}^{2}-R_{a} \end{array}\right]}_{.}$$

According to the new form of (12), Equation (8) can be further rewritten as follows:(15)FD=G.

Therefore, the position matrix in Equation (9) can be provided as follows:(16)$$D={\left[x_{j},y_{j},Q_{j}\right]}^{T}={\left(F^{T}F\right)}^{-1}F^{T}G.$$

Lastly, the estimated coordinates of a normal node *N_j_* are marked as $\left(x_{j}^{*},y_{j}^{*}\right)$ and can be obtained by matrix vectors:(17)$$\left\{\begin{array}{c} x_{j}^{*}=D\left(1\right)\\ y_{j}^{*}=D\left(2\right). \end{array}\right.$$

### 4.4. Initialization of APC

The main parameters of the APC algorithm are initialized in this step, including the population size *m*, maximum iterations *K*, iteration counter *k*, seeding probability *p_s_*, growing probability *p_g_*, fruiting probability *p_f_*, and artificial plant individual *x*.

The artificial plant individual *x* is encoded as a binary string for seeding.
(18)$$x=\{x_{1},x_{2},x_{3},\ldots ,x_{i},\ldots \}.$$

All binary bits {*x_i_*} constitute a feasible solution to the positioning problem.

In nature, the population size of plant communities often increases or decreases with seeding and evolution. Here, we define the seeding rate as representing the ratio of the population size after a seeding operation to the original population size before seeding. The relationship between the seeding rate and the population size of APC is shown in Table 2.

In Table 2, there is a positive relationship between the seeding rate and the population size after 50 iterations. If the seeding rate is less than 1.0, the population size becomes smaller and smaller and the artificial plant community gradually loses its search ability. If the seeding rate is greater than 1.0, the population size becomes larger and larger and the search ability of the APC will continue to increase. However, the convergence speed also declines with the increase in the population size, and the temporal and spatial performance of the algorithm deteriorates sharply on a personal computer. Hence, it is recommended that the population size *m* of the APC is fixed.

The artificial plant individual *x* is encoded into one or more feasible solution variables of the unknown nodes. The positions of the unknown nodes can be encoded by real numbers, and the length of the code can be determined according to the accuracy of the solution. The longer the coding is, the higher the accuracy of the solution is; however, the amount of calculation also becomes correspondingly large. The shorter the coding is, the lower the accuracy of the solution, but a smaller amount of calculation is required.

The target localization calculation function should be defined in the initialization stage of the APC algorithm. The target localization calculation function (cam) selected as the fitness function for the APC algorithm focuses on the minimum positioning error. Here, the proposed fitness function for APC is shown as follows:(19)$$fitness\left(x,y\right)=\min{\left(\sum_{i=1}^{k}\left|\left.\sqrt{{\left(x-x_{i}\right)}^{2}+{\left(y-y_{i}\right)}^{2}}-d_{i}\right|\right.\right)}_{,}$$
where the coordinates of the unknown node are (*x*, *y*), and the coordinates of the neighbor node are (*x_i_*, *y_i_*). The neighbor nodes may be anchor nodes or unknown nodes. The approximate distance *d_i_* of each neighbor node can be obtained from Equation (7).

### 4.5. Seeding of APC

When seeding, the individuals or seeds are generated according to the predetermined population size *m*. An artificial plant individual is randomly selected for seeding, where all its bits are randomly set by a seeding probability of *p_s_*. In the first iteration of the seeding calculation, there is no artificial plant individual and all seeds are randomly produced. In the subsequent iterative calculation, the best solutions in the previous iterative calculation can be selected as seeds according to the seeding probability *p_s_*, and a small number of random seeds are also generated with probability 1 − *p_s_*. The smaller the seeding probability, the stronger the global search capability, but the slower the convergence. On the contrary, the greater the seeding probability, the weaker the global search capability, but the faster the convergence—even premature convergence to the local optimal solution.
(20)$$x=\left\{\begin{array}{l} \mathrm{If}\;\mathrm{in}\;\mathrm{the}\;\mathrm{first}\;\mathrm{interation}\;\mathrm{of}\;\mathrm{seeding}\;\mathrm{operation},\;x=\mathrm{random}()\\ \mathrm{else}\;\mathrm{if}\;p_{s},x=\mathrm{the}\;\mathrm{best}\;\mathrm{fruits}\;\mathrm{in}\;\mathrm{the}\;\mathrm{last}\;\mathrm{iteration}\\ \mathrm{else}\;\mathrm{if}\;1-p_{s},x=\mathrm{random}() \end{array}\right..$$

According to the fitness function in Equation (19), the artificial plant community can implement the first comparison in an iterative computing process and select the best fruits for the next seeds according to the probability *p_s_*. On one hand, the seeding operation keeps the best fruits in the last iteration by *p_s_* and ensures the ability of fast convergence; on the other hand, it also produces some random seeds by 1 − *p_s_* and maintains a certain global search capability.

### 4.6. Growing of APC

Unlike traditional artificial intelligence algorithms, which always use a fixed population size, the population size of the artificial plant community algorithm is variable. After seeding, not all artificial plant individuals or seeds can survive, and the artificial plant community uses a fitness function to judge whether an individual can survive in current environment. Therefore, the population size decreases from *m* in the seeding operation to *p_g_* × *m* in the growing operation. Artificial plant individuals with high fitness function values often have better survival conditions and survival probabilities than other individuals. In the growing operation, the fitness function can help the artificial plant individuals to select the best solutions for the next fruiting operation. The scoring process of the growing operation can help the artificial plant community to converge to a feasible solution.

According to the fitness function in Equation (19), the artificial plant community can implement the second comparison in an iterative computing process and obtain the score to evolve. Only the individuals with high fitness can survive, while the individuals with low fitness die. Only *p_g_* percent of the artificial plant individuals that survive the seeding and growing operations can carry out the fruiting operation in the next step and produce new individuals.
(21)$$x=\left\{\begin{array}{l} \mathrm{if}\;\mathrm{the}\;\mathrm{fitness}\;\mathrm{is}\;\mathrm{high}\;\mathrm{by}\;p_{g},x=\mathrm{the}\;\mathrm{best}\;\mathrm{seeds}\;\mathrm{in}\;\mathrm{the}\;\mathrm{last}\;\mathrm{iteration}\\ \mathrm{else}\;\mathrm{if}\;\mathrm{the}\;\mathrm{fitness}\;\mathrm{is}\;\mathrm{low}\;\mathrm{by}\;1-p_{g},x=\mathrm{zero} \end{array}\right..$$

In every iterative computation, all artificial plant individuals or seeds are ranked from high to low according to the results calculated by the fitness function in Equation (19); the best individuals or seeds are selected for growth according to the growing probability *p_g_*, but the remaining 1 − *p_g_* individuals or seeds will die and are set to zero. Therefore, the artificial plant community can respond to the evaluation results of environmental fitness and the population size decreases. The smaller the growing probability *p_g_*, the stronger the global search capability, but the slower the convergence. On the contrary, the greater the growing probability *p_g_*, the weaker the global search capability, but the faster the convergence—even premature convergence to the local optimal solution. After the growing operation, the artificial plant community can keep the best individuals or seeds according to the fitness comparison and reduce the risk of falling into the local optimal solution.

### 4.7. Fruiting of APC

During fruiting, the population size recovers from *p_g_* × *m* to *m*, and the individuals with higher fitness can produce more fruits. The optimal solution with the highest fitness can be obtained as $x*=\{x_{1}*,x_{2}*,x_{3}*,\ldots ,x_{i}*,x_{i+1}*,\ldots ,x_{n}*\}$, which produces an identical seed through parthenogenesis and keeps it intact for the next iteration. In the fruiting operation, every artificial plant individual is allowed to be fruit, and the population size of the artificial plant community can be restored. The artificial plant individuals with higher fitness can fruit more, which helps the community to retain more high-quality individuals.

According to the fitness function in Equation (19), the artificial plant community can implement the third comparison in an iterative computation and search for the best solutions for fruiting. Natural plant individuals often require the help of additional conditions to implement pollination or fruit-bearing, such as bees, butterflies, wind, water, etc. However, these conditions are not mandatory here, and they are replaced by the fruiting probability *p_f_*. The same individual is allowed to pair with different individuals, and the individuals with higher fitness can pair more frequently. Through the fruiting operation, many new individuals are produced, combining the parents’ features. The fruiting operation chooses several aspects from the parents according to the fruiting probability *p_f_*, then recombines other elements with the probability of 1 − *p_f_* to produce a new offspring. A single artificial parent can also produce a new artificial child by the fruiting probability *p_f_*, and randomly produce the remaining part by the probability of 1 − *p_f_*, to simulate the parthenogenesis in a natural plant community. For the non-optimal artificial plant parents $x=\{x_{1},x_{2},x_{3},\ldots ,x_{i},x_{i+1}\ldots ,x_{n}\}$, $y=\{y_{1},y_{2},y_{3},\ldots ,y_{i},y_{i+1}\ldots ,y_{n}\}$, and $z=\{z_{1},z_{2},z_{3},\ldots ,z_{i},z_{i+1}\ldots ,z_{n}\}$, their fruiting individuals *x*’ can be obtained as follows, where *i* corresponds to the position of probability *p_f_*:(22)$$x=\left\{\begin{array}{l} \mathrm{if}\;x\;\mathrm{is}\;\mathrm{the}\;\mathrm{optimal}\;x*,x’=x*\\ \mathrm{else}\;\mathrm{if}\;\mathrm{the}\;\mathrm{population}\;\mathrm{size}\;\mathrm{is}\;\mathrm{less}\;\mathrm{than}\;p_{g}\;\mathrm{of}\;m,x’=\{x_{1},x_{2},x_{3},\ldots ,x_{i},y_{i+1},\ldots ,y_{n}\}\\ \mathrm{else}\;\mathrm{if}\;\mathrm{the}\;\mathrm{population}\;\mathrm{size}\;\mathrm{is}\;\mathrm{less}\;\mathrm{than}\;m,x’=\{x_{1},x_{2},x_{3},\ldots ,x_{i},z_{i+1},\ldots ,z_{n}\} \end{array}\right..$$

As we can be seen from Equation (22), some of the characteristics of this process differ from traditional artificial intelligence algorithms. First, the best individual is retained in the fruiting operation and can be used as a seed for the next iterative computation. Second, after growing with the highest fitness, the best individuals can cross and produce *p_g_* × *m* of new fruits or individuals for the next seeding operation. Third, for the remainder (1 − *p_g_*) × *m* of new individuals, they can again be generated by crossing the fruiting operations of the best individuals. After the fruiting operation, a new generation of APCs is obtained with the same population size *m*, and they are ranked according to Equation (19). Therefore, the optimal individuals in the growing operation are retained in the fruiting operation, and they can cross to search for better, more feasible solutions.

The fruiting probability *p_f_* determines the ratio of information of high-quality individuals. The higher the fruiting probability *p_f_*, the more information is retained, and the smaller the generation gap is. A high fruiting probability *p_f_* is helpful for retaining high-quality individuals and decreasing the convergence rate. On the contrary, a small fruiting probability *p_f_* retains fewer of these individuals and the generation gap is smaller, since there is either a higher level of information exchange between two individuals or more new information.

If the calculation is not finished, the best solutions after the fruiting operation are selected as seeds for the next iterative computation. Otherwise, the optimal solution calculated by the fruiting operation is output as the optimal solution $\left(x_{j}^{*},y_{j}^{*}\right)$ to the positioning problem in WSNs.

### 4.8. End Judgment

The proposed APC algorithm uses an iterative calculation to simulate the process of the plant community repeatedly seeking the optimal solution. The artificial plant individuals with higher fitness values have a higher survival probability and are more likely to be the optimal solution. Through three main operations, seeding, growing, and fruiting, it must be determined whether the solutions to the positioning problem in WSNs meet the end requirements and whether the calculation can be stopped. The judging conditions before an optimal solution can be produced include a predefined accuracy, or computing error, or the maximum of iterations, or the total computational time.

Assume that the optimal solution obtained by solving the coordinates of the unknown node *N_j_* is $\left(x_{j}^{*},y_{j}^{*}\right)$, and the actual coordinates are $\left(x_{j}^{\#},y_{j}^{\#}\right)$. For a WSN with *n* nodes and the communication range of *r*, the accuracy of the positioning algorithm can be calculated as follows:(23)$$Accuracy=1-\frac{\sum_{j=1}^{n}\sqrt{{\left(x_{j}^{\#}-x_{j}^{*}\right)}^{2}+{\left(y_{j}^{\#}-y_{j}^{*}\right)}^{2}}}{n\times r}.$$

The greater the accuracy value calculated by Equation (23), the better it is, and the more suitable it is for output as a feasible solution.

The positioning error *e* can be calculated as follows:(24)$$e=\sqrt{{\left(x_{j}^{\#}-x_{j}^{*}\right)}^{2}+{\left(y_{j}^{\#}-y_{j}^{*}\right)}^{2}}.$$

If the accuracy computed by Equation (23) fits the requirement or the error *e* calculated by Equation (24) is less than the preset error threshold *e_th_*, the calculation is completed and the optimal solution $\left(x_{j}^{*},y_{j}^{*}\right)$ is output as the final position of node *N_j_*. Otherwise, the optimal solution is taken as the seeds returning to the seeding operation for the next iterative calculation.

### 4.9. Algorithm Flow of APC

The APC algorithm flow simulates the evolution mechanism of a nerveless plant community, as shown in Figure 2. Based on the repeating operations of seeding, growing, and fruiting, the APC algorithm can use swarm learning and evolutionary mechanisms to help to efficiently solve the positioning problem in WSNs.

The whole algorithm flow is composed of two main steps and a large circulation.

The first main step, detailed above, is a weighted DV-Hop algorithm and includes computing the minimum hop, calculating the average hop size, and estimating the positions of unknown nodes. The main aim of these steps is to obtain the minimum number *h_ij_*_min_ of hops to all anchor nodes *A*, the weighted average hop size $Avgh_{j}^{’}$, and the coordinates $\left(x_{j}^{*},y_{j}^{*}\right)$ of a normal node *N_j_*.

The next major step is an APC optimization algorithm which consists of the initialization of APC, the seeding of APC, the growing of APC, the fruiting of APC, and an end judgment. The stage includes a large circulation, where the artificial plant community repeatedly searches for the optimal solution to the positioning problem in WSNs.

### 4.10. Pseudo-Code of APC

According to the APC algorithm flow in Figure 2, the corresponding pseudo-code is shown in Figure 3.

The first main step is shown in Lines 1~26 of Figure 1. Lines 1~4 work to obtain the parameters of the network and Line 5 produces the original topology of the WSN. Then, Lines 6 and 7 initialize the hop counter and distance counter, and Lines 8~16 update the hop counter and distance counter. After that, Lines 17~25 calculate the weighted average hop size, and then Line 26 estimates the unknown node position using Equation (17).

The second major step is shown in Lines 27~41 of Figure 1. Lines 27~31 initialize the parameters of an artificial plant community. In iterative computing, the three learning factors *p_s_*, *p_g_*, and *p_f_* help the APC to determine the optimal position. In Lines 32~40, there are *K* iterations of circulation, and the artificial plant community repeatedly searches for the optimal solution by seeding, growing, and fruiting. The fitness function is calculated three times in Lines 33, 35, and 37, and the justification of the end condition is shown in Line 39. In each iteration, the best solutions for the node position are recorded for the next round of comparison. Finally, the best positions of the unknown nodes are output in Line 41.

The best solutions in each step are compared according to the fitness function in Equation (19) or other forms before the optimal solution is output. If the end condition is satisfied, such as the accuracy in Equation (23) and the error in Equation (24), then the optimal solution is output and the whole algorithm is stopped. When the number of iterations reaches the maximum value, or the limited computing time is used up, the optimal solution is also output. If the end conditions are not satisfied, the APC uses the latest solutions to seed, and a new round of calculations begins.

## 5. Experimental Results and Discussion

In this section, the experimental results are outlined and a performance comparison of the proposed algorithms and other artificial intelligence methods are offered. All algorithms are implemented in the MATLAB simulator on a personal computer, and do not require a professional biological laboratory or biological experimental operation. The experimental platform includes AMD Ryzen 3 4300U with Radeon Graphics 2.70 GHz CPU, 8.00 GB RAM, 64-bit Windows 10 operating system, and MATLAB R2018a simulation software. MATLAB R2018a is a popular simulation software and a strong matrix computing platform used to develop algorithms and analyze data on a personal computer. A series of experiments are designed to evaluate the position accuracy and computing performance of the proposed algorithms in static WSNs. Additionally, related artificial intelligence algorithms include Particle Swarm Optimization (PSO) [4,35,36], the Artificial Bee Colony (ABC) [8,29], Convolutional Neural Networks (CNN) [18,19], Fuzzy Logic (FL) [22,34], Ant Colony Optimization (ACO) [22,37], the Genetic Algorithm (GA) [23,38], and the Artificial Fish Swarm Algorithm (AFSA) [24].

Three main metrics are compared in the following experiments, including the positioning accuracy, position error per node, and computing time. Some varying parameters are evaluated, including the nodes’ communication range, the total number of sensor nodes, the total number of anchor nodes in random distribution topologies, and the population size of the corresponding algorithm. It is assumed that the optimal solution obtained by solving the coordinates of the unknown node *N_j_* is $\left(x_{j}^{*},y_{j}^{*}\right)$, and the actual coordinates are $\left(x_{j}^{\#},y_{j}^{\#}\right)$. The position accuracy is expressed as the average error per communication range between the coordinates $\left(x_{j}^{*},y_{j}^{*}\right)$ and $\left(x_{j}^{\#},y_{j}^{\#}\right)$, which can be used to measure the superiority of an algorithm. The position error is expressed as the deviation between the estimated coordinates $\left(x_{j}^{*},y_{j}^{*}\right)$ and the exact coordinates $\left(x_{j}^{\#},y_{j}^{\#}\right)$. The accuracy is calculated using Equation (23), and then the positioning error *e* is calculated by the difference between the fitness function values of the last two iterations.

For swarm intelligence algorithm, different population size or different parameters will affect the solution accuracy or speed. This paper mainly illustrates the applicability of this algorithm. The population size of 20 is feasible for most AI algorithms and easy to compare. For the sake of fairness, the simulation parameters in the different algorithms are unified as much as possible. Only 20 plant individuals (*m* = 20) are used in our algorithm, and the simulation parameters used in the different algorithms remain unchanged in every test. The PSO [4,35,36] sets the population size *m* = 20, the location limitation as 0.5, the speed limitation as [−0.5, 0.5], the self-learning factor as c1 = 1.5, and the social learning factor as c2 = 1.5. The ABC [8,29] sets the searching capability limit = 100, the population size *m* = 20, and the neighbor size NI = 10. The parameters of the CNN [18,19] are set as a convolutional neural network with three convolution cores, eight input channels (c_in_ = 3), and eight output channels (c_out_ = 3). The learning rate of the offset item is twice that of the weight. The extension edge is set to 0, the weight is initialized to Gaussian, and the value of the constant offset item is always 0. The parameters of FL [22,34] are set as *m* = 20 classes, five fuzzy linguistic valuables, {NB, NS, Z, PS, PM}, namely {negative big, negative small, zero, positive small, positive big}, and use Mamdani fuzzy reasoning. The parameters of ACO [22,37] are set as the pheromone importance = 1.0, the importance of heuristic factors = 5.0, the pheromone volatilization factor = 0.1, and *m* = 20 ants. The GA [23,38] sets the population size *m* = 20, the chromosome length Lind = 20, the crossover probability px = 0.7, and the mutation probability pm = 0.01. AFSA [24] sets the fish number *m* = 20, the maximum number of attempts = 100, the perceptual distance = 1.5, the congestion factor = 0.5, and the fish step = 0.2.

### 5.1. Algorithm Comparison in Terms of Positioning Accuracy

In this section, simulation experiments are performed to survey the positioning accuracy of the proposed algorithm and other position algorithms in random wireless sensor networks with a communication range *r*. The generated WSN topology is in an area of 100 m × 100 m, which includes a number of sensor nodes with some anchor nodes. All sensor nodes are randomly distributed and have the same communication radius. Figure 4 presents the influence of four indicators on the positioning accuracy in the same network topologies, which is calculated from Equation (23). In this section, several scenarios are considered to discuss the influence of the main parameters, especially the communication range, the total number of sensor nodes, the percentage of anchor nodes, and the population size, on the accuracy of the proposed algorithm and other position algorithms. In this series of scenarios, the topology is based on the random deployment of nodes with an empty area in WSNs.

Figure 4a shows the scenario of the communication range vs. the positioning accuracy (200 sensor nodes, 20% anchor nodes, population size *m* = 20). Here, the communication range *r* varies from 10 m to 40 m, and the increase affects the positioning accuracy. As we can see from Figure 4a, the positioning accuracy of all algorithms tends to increase with increasing communication range because there are more connections. Furthermore, it can be noted that the proposed algorithm has higher positioning accuracy, especially when the communication range is 20 m. The proposed method outperforms the other algorithms in terms of positioning accuracy.

Figure 4b provides the scenario of the total number of sensor nodes vs. the positioning accuracy (communication range *r* = 20 m, 20% anchor nodes, population size *m* = 20). The total number of sensor nodes varies from 100 to 500, and the percentage of anchor nodes and communication range *r* are fixed at 20% and 20 m, respectively. As can be seen from Figure 4b, the positioning accuracy is increasing with the total number of sensor nodes from 100 to 500 because the total number of anchor nodes of the WSN is increasing, and there are more connections between more nodes deployed in the same area. Furthermore, higher positioning accuracy is achieved by the proposed method in approximately *n* = 300, but most other algorithms achieve positioning accuracy below 0.75. Therefore, the proposed algorithm shows better positioning accuracy and stability as the number of anchor nodes increases.

Figure 4c shows the scenario of the percentage of anchor nodes vs. the positioning accuracy (communication range *r* = 20 m, 200 sensor nodes, population size *m* = 20). The percentage of anchor nodes varies from 10% to 40%, but the total number of sensor nodes and the communication range *r* are fixed at 200 and 20 m, respectively. In this scenario, it can be seen that both the proposed algorithm and CNN [18,19] algorithms achieved a higher positioning accuracy versus the percentage of anchor nodes compared with the other algorithms. Although all algorithms tend to improve with the increasing percentage of anchor nodes, the proposed algorithm outperforms other algorithms with higher positioning accuracy and does not require a high density of anchor nodes.

Figure 4d displays the scenario of the population size vs. positioning accuracy (communication range *r* = 20 m, 200 sensor nodes, 20% anchor nodes). The percentage of anchor nodes, the total number of sensor nodes, and the communication range *r* are fixed at 20%, 200, and 20 m, respectively. The population size *m* ranges from 10 to 50. It can be seen from Figure 4d that all algorithms achieve smaller growth in positioning accuracy versus the population size as compared with the increasing communication range in Figure 4a and the increasing percentage of anchor nodes in Figure 4c. As can be observed, different population sizes can affect the solution’s accuracy or speed of swarm intelligence algorithms. A population size of 20 is not sufficiently large for many AI algorithms; however, it is enough for us to test the feasibility of our APC algorithm. Nevertheless, the proposed algorithm still achieves satisfactory accuracy.

Table 3 shows a comparison of the node positioning data statistics between the proposed algorithms and the other algorithms in terms of the minimum, maximum, and average position error. Therefore, the proposed APC algorithm is suitable for use as an accurate positioning method with high accuracy.

### 5.2. Algorithm Comparison in Terms of the Iterative Error

In this section, simulation experiments are performed to survey the iterative error of the proposed algorithm and other position algorithms in random wireless sensor networks with a communication range *r*. The generated WSN topology is in an area of 100 m × 100 m, which includes a number of sensor nodes with some anchor nodes. All of the sensor nodes are randomly distributed and have the same communication radius. Figure 5 shows the influence of four indicators on the iterative error in the same network topologies, which is calculated by the error between the last two iterations. Several scenarios are considered in this section to discuss the influence of the main parameters, especially the communication range, the total number of sensor nodes, the percentage of anchor nodes, and the population size, on the error of the proposed algorithm and other position algorithms. In this series of scenarios, the topology is based on the random deployment of nodes with an empty area in WSNs.

Figure 5a shows the scenario of communication range vs. iterative error (200 sensor nodes, 20% anchor nodes, population size *m* = 20). Here, the communication range *r* varies from 10 m to 40 m, and the increase does not greatly affect the iterative error. As can be seen from Figure 5a, the iterative errors of all algorithms tend to remain stable as the communication range increases since there is less of an effect on the iterative error. Furthermore, it can be noted that the proposed algorithm has a lower iterative error.

Figure 5b provides the scenario of the total number of sensor nodes vs. the iterative error (communication range *r* = 20 m, 20% anchor nodes, population size *m* = 20). The total number of sensor nodes varies from 100 to 500, and the percentage of anchor nodes and the communication range *r* are fixed at 20% and 20 m, respectively. As can be seen from Figure 5b, the iterative error decreases as the total number of sensor nodes ranges from 100 to 500 because the total number of anchors increases and the possible connections increase. In comparison with the proposed algorithm, it is observed that the iterative error is below ±0.01%. Therefore, the performance of the proposed algorithm is comparable with those of the other positioning algorithms.

Figure 5c shows the scenario of the percentage of anchor nodes vs. the iterative error (communication range *r* = 20 m, 200 sensor nodes, population size *m* = 20). The percentage of anchor nodes varies from 10% to 40%, but the total number of sensor nodes and communication range *r* are fixed at 200 and 20 m, respectively. In this scenario, the iterative error of all the algorithms tends to decrease with the increase in the percentage of anchor nodes because more anchors may help to position unknown nodes. The proposed algorithm achieved a lower iterative error versus the percentage of anchor nodes as compared with the other algorithms; it also shows better stability with the increasing number of anchor nodes. The APC algorithm does not require a high density of anchor nodes and can be used in low-density anchor networks.

Figure 5d displays the scenario of the population size vs. the iterative error (communication range *r* = 20 m, 200 sensor nodes, 20% anchor nodes). The percentage of anchor nodes, the total number of sensor nodes, and communication range *r* are fixed at 20%, 200, and 20 m, respectively. The population size *m* varies from 10 to 50. It can be seen from Figure 5d that the iterative error of all the algorithms tends to decrease with the increase in the population size because a larger population size may help to position unknown nodes. The proposed algorithm can achieve a lower iterative error versus the population size as compared with the other algorithms.

Table 4 shows a comparison of the node positioning data statistics between the proposed algorithms and the other algorithms in terms of the minimum, maximum, and average positioning error, respectively. Hence, the proposed APC algorithm is suitable for use as an accurate positioning method with a low iterative error.

### 5.3. Algorithm Comparison with Regard to Computing Time

In this section, simulation experiments are performed to survey the computing time of the proposed algorithm and other position algorithms in random wireless sensor networks with a communication range *r*. The generated WSN topology is in an area of 100 m × 100 m, which includes a number of sensor nodes with some anchor nodes. All sensor nodes are randomly distributed and have the same communication radius. Figure 6 presents the influence of four indicators on the computing time in the same network topologies on the same personal computer. In this section, several scenarios are considered to discuss the influence of the main parameters, especially the communication range, the total number of sensor nodes, the percentage of anchor nodes, and the population size, on the computing time of the proposed algorithm and other position algorithms. In this series of scenarios, the topology is based on the random deployment of nodes with an empty area in WSNs.

Figure 6a shows the scenario of the communication range vs. computing time (200 sensor nodes, 20% anchor nodes, population size *m* = 20). Here, the communication range *r* varies from 10 m to 40 m, and the increase affects the computing time. As can be seen from Figure 6a, the positioning computing time of all the algorithms tends to increase with the increasing communication range because there are more connections and more complex topologies. Furthermore, it can be noted that the proposed algorithm has shorter computing times, but the computing time of CNN [18,19] is too costly. At the same positioning accuracy, the proposed method outperforms the CNN [18,19] algorithm and other algorithms in terms of computing time.

Figure 6b provides the scenario of the total number of sensor nodes vs. the computing time (communication range *r* = 20 m, 20% anchor nodes, population size *m* = 20). The total number of sensor nodes varies from 100 to 500, and the percentage of anchor nodes and communication range *r* are fixed at 20% and 20 m, respectively. As can be seen from Figure 6b, the computing time of all algorithms increases with the total number of sensor nodes because of the increasing node density and the presence of more connections between nodes of the WSN in the same area. Furthermore, shorter computing times are achieved by the proposed method; however, among all of the algorithms considered, the CNN [18,19] algorithm requires the longest computing times. In comparison, it is observed that the computing time of heuristic algorithms is below 1.5. Therefore, the proposed algorithm outperforms all the other positioning algorithms and shows a better computing time and searching stability with the increasing number of anchor nodes.

Figure 6c shows the scenario of the percentage of anchor nodes vs. the computing time (communication range *r* = 20 m, 200 sensor nodes, population size *m* = 20). The percentage of anchor nodes varies from 10% to 40%, but the total number of sensor nodes and communication range *r* are fixed at 200 and 20 m, respectively. In this scenario, it can be seen that all the algorithms achieved shorter computing times versus the percentage of anchor nodes, but the CNN [18,19] algorithm requires longer computing times compared with the other algorithms. The proposed algorithm outperforms the CNN [18,19] algorithms, with computing times approximately three times lower.

Figure 6d displays the scenario of the population size vs. the computing time (communication range *r* = 20 m, 200 sensor nodes, 20% anchor nodes). The percentage of anchor nodes, the total number of sensor nodes, and the communication range *r* are fixed at 20%, 200, and 20 m, respectively. The population size *m* varies from 10 to 50. It can be seen from Figure 6d that both the proposed algorithm and other heuristic algorithms can achieve an increased computing time versus the population size, which means longer convergence times for solving the positioning problem. In contrast, the computing performance of the CNN [18,19] algorithm deteriorates sharply with the increase in convolution kernels, and it is difficult even to complete the positioning task on a personal computer.

Table 5 shows a comparison of the data statistics between the proposed algorithm and the other algorithms in terms of the minimum, maximum, and average computing times. Therefore, the proposed APC algorithm is suitable for use as a fast positioning method in large-scale WSNs with shorter computing times.

### 5.4. Theoretical Performance Analysis and Advantages

This section further explores the underlying reasons behind the experimental results in the previous sections. The theoretical analysis of time and space performance is shown in Table 6. Related algorithms include the proposed APC algorithm, Particle Swarm Optimization (PSO) [4,35,36], Artificial Bee Colony (ABC) [8,29], Convolutional Neural Networks (CNN) [18,19], Fuzzy Logic (FL) [22,34], Ant Colony Optimization (ACO) [22,37], the Genetic Algorithm (GA) [23,38], the Artificial Fish Swarm Algorithm (AFSA) [24], Simulated Annealing (SA) [25], Grey Wolf Optimization (GWO) [31], the Firefly Algorithm (FA) [31], the Sparrow Search Algorithm (SSA)[42], the Coot Bird Algorithm (CBA) [43], the Fruit Fly Optimization Algorithm (FOA) [44], and the Whale Optimization Algorithm (WOA) [45].

For a fair comparison, it is assumed that all algorithms use the same iteration steps *t* to solve the positioning problem in WSNs with *n* nodes. The time performance and space performance of APC are nearly linear and are related to the scale *n* of WSNs, the population size *m* of the artificial plant community, and the number *t* of iterations.

From Table 6, it can be seen that the time performance and space performance of the APC, ABC [8,29], FL [22,34], CNN [18,19], PSO [4,35,36], ACO [22,37], GA [23,38], AFSA [24], GWO [31], FA [31], SSA [42], CBA [43], FOA [44], and WOA [45] are related to the scale *n* of the WSNs, the number *t* of iterations, and the population size *m* of the artificial intelligence swarm. All of these heuristic algorithms are better than CNN [18,19] in terms of time performance and space performance, but CNN may achieve better accuracy. The time performance and space performance of the CNN [18,19] are related to the scale *n* of the WSNs, the population size *m* of convolution cores, the number *c_in_* of input channels, and the number *c_out_* of output channels. However, there are no artificial individuals or swarm learning in SA [25] and its performance depends on the search strategies.

For a WSN with *n* nodes and the same iteration steps *t*, different iterative algorithms use different artificial individuals with the same number of *m* to search for the optimal positioning solutions, such as the artificial plant individuals (APC), particles or birds (PSO [4,35,36]), artificial bees (ABC [8,29]), artificial neurons (CNN [18,19]), fuzzy classes (FL [22,34]), artificial ants (ACO [22,37]), chromosomes (GA [23,38]), artificial fish (AFSA [24]), grey wolves (GWO [31]), fireflies (FA [31]), sparrows (SSA [42]), coot birds (CBA [43]), fruit flies (FOA [44]), and whales (WOA [45]). The computing performance of CNN [18,19] is strongly related to network structure and is independent of time.

Furthermore, a theoretical comparison of the positioning accuracy between the APC and related algorithms is presented Table 7. The main comparison factors for positioning accuracy include variable population size, optimal solution preserving, global searching capability, and fitness comparison per iteration.

Compared with traditional artificial intelligence algorithms, APC has some advantages.

First, the variable population size accelerates the convergence performance. Only individuals with high fitness can survive the growing operation, while individuals with low fitness die. During fruiting, the population size recovers, and the individuals with higher fitness can produce more fruits.

Second, the optimal solution in each iterative computation can be well preserved as a parthenogenesis fruit for the next seeding operation. The optimal solution calculated in each iteration has the highest priority in the APC algorithm. In traditional AI algorithms, the optimal solution is randomly changed and can easily be covered by new generations.

Third, seeding and fruiting operations provide good global searching capabilities. In each seeding operation, a small number of new seeds will be generated through random seeding. In the fruiting operation, the individuals will learn from each other, and the individuals with high fitness can produce a greater number of different fruits which can increase the search ability near the feasible solutions.

Forth, fitness comparison per iteration is conducted three times in the APC algorithm, while it is usually only conducted once in other AI algorithms. The seeding, growing, and fruiting operations can compare fitness per iteration to provide good local searching capabilities. In the seeding operation, the fruits with high fitness can survive and be seeded by a probability *p_s_*, while the fruits with low fitness die. In the growing operation, the seeds with high fitness can survive with a probability *p_g_*, while the seeds with low fitness die. In the fruiting operation, the individuals with high fitness can produce more fruits by a probability of *p_f_*.

Fifth, there are few parameters and the operation is simple. Seeding, growing, and fruiting are simple binary logic operations, which require few computing resources and are quick to execute.

Last, the proposed APC algorithm has low costs, low communication overheads, no additional requirements for equipment, low power consumption, and better positioning accuracy. It can obtain accurate positioning solutions without requiring a high density of anchor nodes or extra hardware.

Unlike traditional artificial intelligence algorithms with a fixed population size and only one fitness comparison per iteration, the population size of the APC algorithm is variable and the fitness comparison is implemented three times per iteration. Regarding the theoretical performance comparison, the APC algorithm can easily obtain accurate solutions and can also be applied in as many areas as mainstream artificial intelligence algorithms; moreover, its time performance and space performance are similar to those of other popular algorithms. The universality of seeding, the selectivity of growing, and the diversity of fruiting effectively balance the algorithm’s global search ability and fast convergence ability. The experimental results of APC, along with the reference algorithms, will be analyzed and surveyed in the next section with reference to additional comparisons.

## 6. Conclusions

In this work, a novel positioning algorithm called artificial plant community, APC, was proposed to improve positioning accuracy in WSNs. The novel APC algorithm provides us with the following implications and insights. First, the variable population size can help us to accelerate the convergence performance. Second, a parthenogenesis fruit can help us to well preserve the optimal solution in each iteration for the next seeding operation. Third, the random search in seeding and fruiting operations can help us to obtain good global searching capabilities. Forth, fitness comparison is repeated three times in each iteration and can help us to determine the optimal solution as soon as possible. Fifth, there are few parameters and the operation is simple. Simulation experiments show that the proposed APC algorithm achieved higher positioning accuracy, a lower iterative error, and shorter computing times than the other algorithms. This evolution mechanism of APC is very simple in theory, but it can not only effectively preserve the optimal solution in each calculation, but also retain a certain global search capability. The proposed APC algorithm is expected to be used to solve positioning problems and other complex problems encountered in wireless sensor networks as widely as other artificial intelligence algorithms.

The advantage of the artificial plant community algorithm is its variable population size, as well as its repeated fitness comparisons. When seeding, the individuals or seeds are generated according to the predetermined population size. During the growing operation, the population size changes little. Only the individuals with high fitness can survive, while the individuals with low fitness die. In fruiting, the population size recovers, and the individuals with higher fitness can produce more fruits. When seeding again, the fruits with high fitness can survive and be seeded, while the fruits with low fitness die, and a small number of new seeds are generated through random seeding. This evolution mechanism is very simple in theory, but it effectively preserves the optimal solution in each step of the calculation, and also retains a certain global search capability. The APC algorithm can obtain accurate positioning solutions without requiring a high density of anchor nodes or extra hardware.

The limitations of our study mainly concern the parallelism of APC, which uses iterative computing on a personal computer and is therefore different from a natural plant community. Real plant communities can produce more seeds and more plant individuals in the evolution process to increase their searching abilities and survival capability. Nevertheless, it is difficult to implement true parallelism on a personal computer since the expansion of the population will worsen the computing performance.

In future work, further experiments will be conducted to evaluate more algorithms and parameters in more complex wireless sensor networks. The proposed APC algorithm will be extended to determine the positions of unknown sensor nodes in 3D topologies and mobile environments.

## Figures and Tables

**Figure 1 sensors-23-02804-f001:**
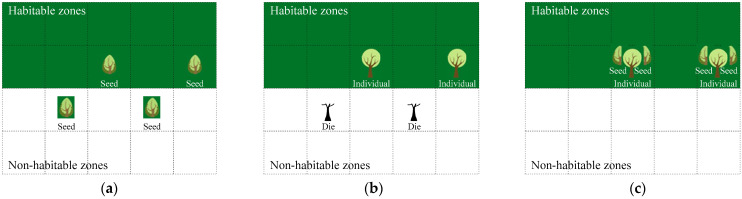
Artificial plant community system [26,49]. (**a**) Seeding; (**b**) growing; (**c**) fruiting.

**Figure 2 sensors-23-02804-f002:**
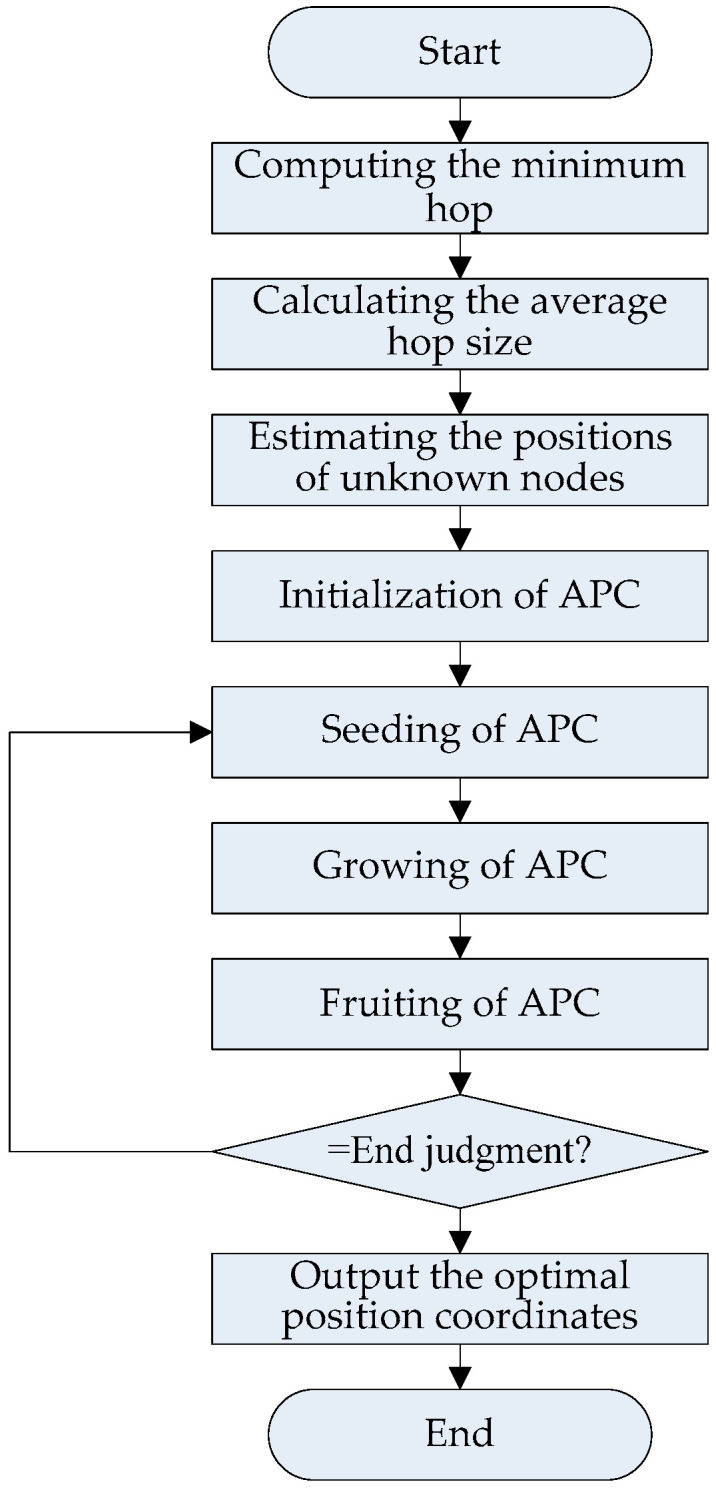
The APC algorithm used to solve the positioning problem in WSNs.

**Figure 3 sensors-23-02804-f003:**
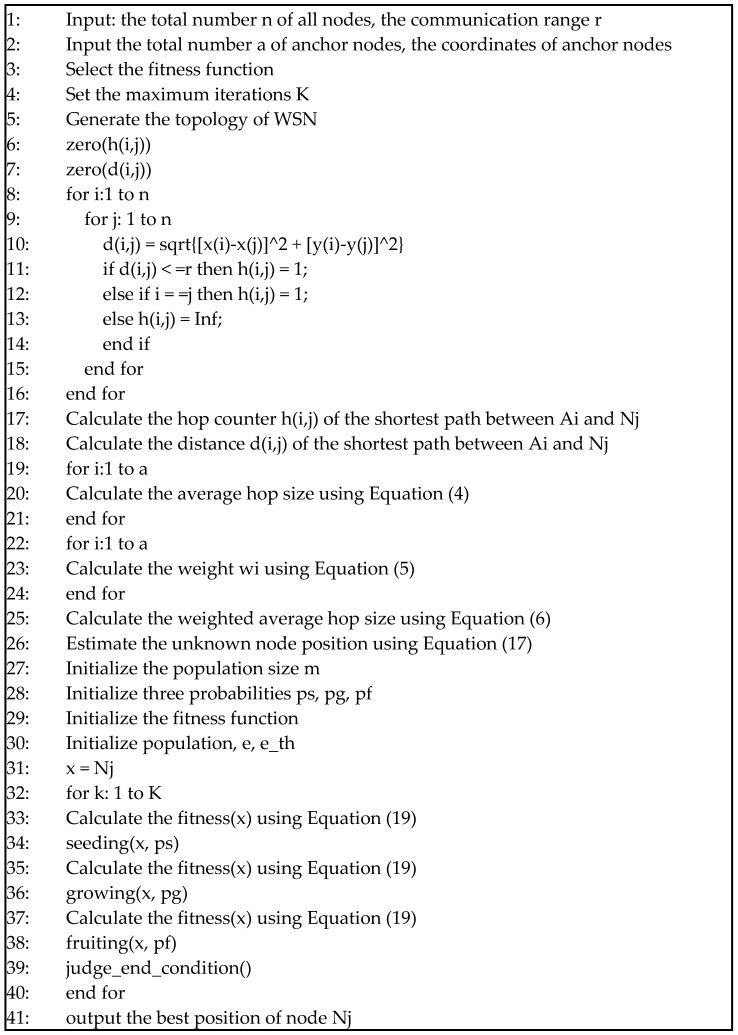
The pseudo-code of APC.

**Figure 4 sensors-23-02804-f004:**
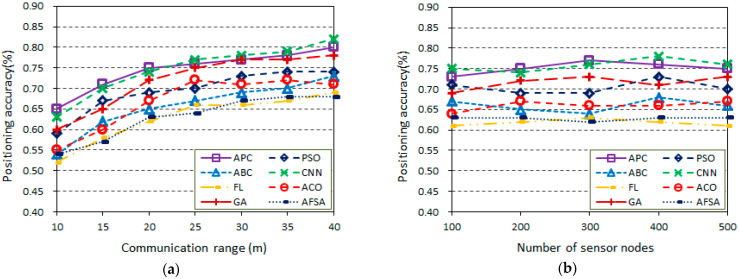
Algorithm comparison in terms of positioning accuracy. (**a**) Communication range vs. positioning accuracy (200 sensor nodes, 20% anchor nodes, population size *m* = 20); (**b**) the total number of sensor nodes vs. positioning accuracy (communication range *r* = 20 m, 20% anchor nodes, population size *m* = 20); (**c**) percentage of anchor nodes vs. positioning accuracy (communication range *r* = 20 m, 200 sensor nodes, population size *m* = 20); (**d**) population size vs. positioning accuracy (communication range *r* = 20 m, 200 sensor nodes, 20% anchor nodes).

**Figure 5 sensors-23-02804-f005:**
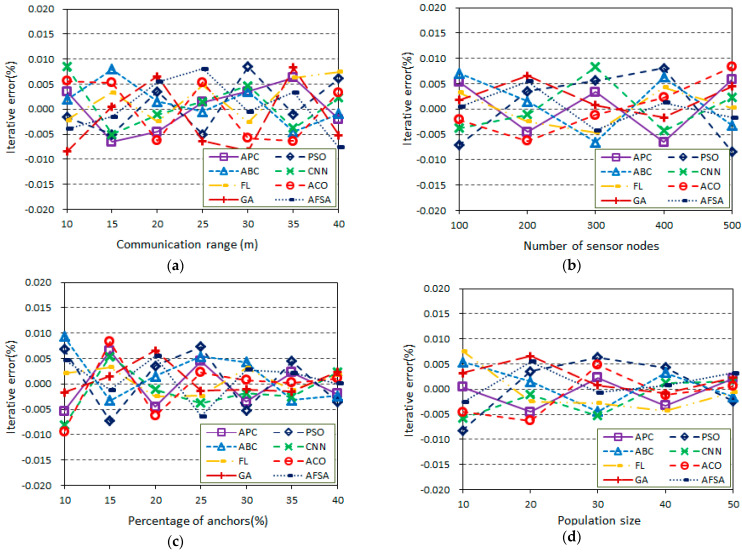
Algorithm comparison in terms of the iterative error. (**a**) Communication range vs. iterative error (200 sensor nodes, 20% anchor nodes, population size *m* = 20); (**b**) the total number of sensor nodes vs. the iterative error (communication range *r* = 20 m, 20% anchor nodes, population size *m* = 20); (**c**) percentage of anchor nodes vs. the iterative error (communication range *r* = 20 m, 200 sensor nodes, population size *m* = 20); (**d**) population size vs. the iterative error (communication range *r* = 20 m, 200 sensor nodes, 20% anchor nodes).

**Figure 6 sensors-23-02804-f006:**
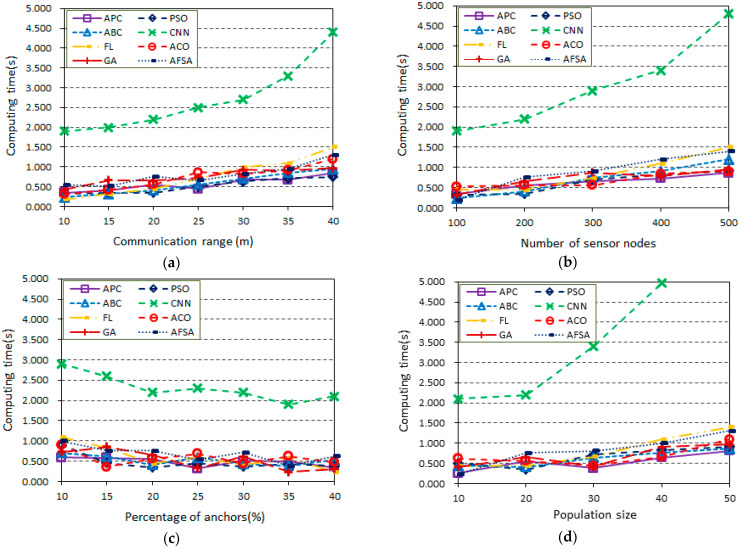
Algorithm comparison in relation to computing time. (**a**) Communication range vs. positioning accuracy (200 sensor nodes, 20% anchor nodes, population size *m* = 20); (**b**) the total number of sensor nodes vs. positioning accuracy (communication range *r* = 20 m, 20% anchor nodes, population size *m* = 20); (**c**) percentage of anchor nodes vs. positioning accuracy (communication range *r* = 20 m, 200 sensor nodes, population size *m* = 20); (**d**) population size vs. positioning accuracy (communication range *r* = 20 m, 200 sensor nodes, 20% anchor nodes).

**Table 1 sensors-23-02804-t001:** Symbol definitions.

Symbol	Definition
*N*	A sensor node set
*N_i_*	A sensor node
*n*	The total number of all nodes
*r*	The communication range
*A*	An anchor node set
*A_i_*	An anchor node
*a*	The total number of anchor nodes
*U*	An unknown node set
*U_i_*	An unknown node
*h_i_* * _j_ *	A hop counter between nodes *N_i_* and *N_j_*
*h_i_* * _j_ * _min_	The minimum value of the hop counter between nodes *N_i_* and *N_j_*
*d_i_* * _j_ *	The estimated distance between nodes *N_i_* and *N_j_*
*avgh_i_*	The average value of the hop counter of node *N_i_*
*s_i_*	The statistical error of node *N_i_*
(*x_i_*, *y_i_*)	The coordinates of node *N_i_*
*K*	The maximum iterations
*k*	An iteration counter
*m*	The population size
*x*	An artificial plant individual
*p_s_*	The seeding probability
*p_g_*	The growing probability
*p_f_*	The fruiting probability
*fitness*	The fitness function
*e*	The positioning error
*e_th_*	The error threshold

**Table 2 sensors-23-02804-t002:** Seeding rate and population size.

Seeding Rate	Population Size after 50 Iterations
0.8	0.8^50^ × original size
1.0	1 × original size
1.2	1.2^50^ × original size
1.5	1.5^50^ × original size
2.0	2^50^ × original size
5.0	5^50^ × original size

**Table 3 sensors-23-02804-t003:** Node positioning data statistics on positioning accuracy (200 sensor nodes, communication range *r* = 20 m, 20% anchor nodes, population size *m* = 20).

Algorithm	Min	Max	Mean
Proposed APC	0.538	0.873	0.747
PSO [4,35,36]	0.543	0.864	0.686
ABC [8,29]	0.527	0.841	0.652
CNN [18,19]	0.554	0.887	0.738
FL [22,34]	0.530	0.838	0.623
ACO [22,37]	0.532	0.856	0.667
GA [23,38]	0.551	0.860	0.724
AFSA [24]	0.536	0.852	0.629

**Table 4 sensors-23-02804-t004:** Node positioning data statistics on the iterative error (200 sensor nodes, communication range *r* = 20 m, 20% anchor nodes, population size *m* = 20).

Algorithms	Min(Abs)	Max(Abs)	Mean
Proposed APC	−0.0005%	0.0087%	−0.00032%
PSO [4,35,36]	0.0006%	−0.0093%	0.00027%
ABC [8,29]	−0.0007%	−0.0095%	0.00065%
CNN [18,19]	0.0003%	−0.0081%	−0.00011%
FL [22,34]	−0.0009%	0.0096%	−0.00064%
ACO [22,37]	0.0006%	0.0091%	−0.00043%
GA [23,38]	0.0005%	−0.0089%	0.00026%
AFSA [24]	−0.0008%	0.0097%	0.00055%

**Table 5 sensors-23-02804-t005:** Data statistics related to computing time (200 sensor nodes, communication range *r* = 20 m, 20% anchor nodes, population size *m* = 20).

Algorithms	Min	Max	Mean
Proposed APC	0.325 s	0.706 s	0.548 s
PSO [4,35,36]	0.351 s	0.812 s	0.545 s
ABC [8,29]	0.386 s	0.909 s	0.647 s
CNN [18,19]	1.837 s	3.104 s	2.262 s
FL [22,34]	0.408 s	1.125 s	0.673 s
ACO [22,37]	0.349 s	0.743 s	0.561 s
GA [23,38]	0.322 s	0.718 s	0.554 s
AFSA [24]	0.410 s	0.981 s	0.756 s

**Table 6 sensors-23-02804-t006:** Theoretical comparison of time and space performance.

Algorithms	Time Performance	Space Performance
Proposed APC	O(nmt)	O(nm)
PSO [4,35,36]	O(nmt)	O(nm)
ABC [8,29]	O(nmt)	O(nm)
CNN [18,19]	$O(n^{2}m^{2}c_{in}c_{out})$	$O(n^{2}c_{in}c_{out})$
FL [22,34]	O(nmt)	O(nm)
ACO [22,37]	O(nmt)	O(nm)
GA [23,38]	O(nmt)	O(nm)
AFSA [24]	O(nmt)	O(nm)
SA [25]	O(nt)	O(n)
GWO [31]	O(nmt)	O(nm)
FA [31]	O(nmt)	O(nm)
SSA [42]	O(nmt)	O(nm)
CBA [43]	O(nmt)	O(nm)
FFO [44]	O(nmt)	O(nm)
WOA [45]	O(nmt)	O(nm)

**Table 7 sensors-23-02804-t007:** Theoretical comparison for positioning accuracy.

Algorithms	Variable Population Size	Optimal Solution Preserving	Global Searching Capability	Fitness Comparison per Iteration
Proposed APC	Reduction—recovery	Parthenogenesis	Artificial plant individuals	Three times
PSO [4,35,36]	Fixed	Randomly changed	Particles or birds	Usually once
ABC [8,29]	Fixed	Randomly changed	Artificial bees	Usually once
CNN [18,19]	Fixed	Randomly changed	Artificial neurons	Usually once
FL [22,34]	Fixed	Randomly changed	Fuzzy classes	Usually once
ACO [22,37]	Fixed	Randomly changed	Artificial ants	Usually once
GA [23,38]	Fixed	Randomly changed	Chromosomes	Usually once
AFSA [24]	Fixed	Randomly changed	Artificial fish	Usually once
SA [25]	Null	Null	Null	Usually once
GWO [31]	Fixed	Randomly changed	Grey wolves	Usually once
FA [31]	Fixed	Randomly changed	Fireflies	Usually once
SSA [42]	Fixed	Randomly changed	Sparrows	Usually once
CBA [43]	Fixed	Randomly changed	Coot birds	Usually once
FFO [44]	Fixed	Randomly changed	Fruit flies	Usually once
WOA [24]	Fixed	Randomly changed	Whales	Usually once

## Data Availability

Not applicable.

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
