# Peer review of "An Artificial Plant Community Algorithm for the Accurate Range-Free Positioning of Wireless Sensor Networks"

_sensors, 2023, doi:10.3390/s23052804_

Round 1

Reviewer 1 Report

Comments:

In order to motivate more research on localization in Wireless Sensor Networks (WSNs), an artificial plant community algorithm for accurate range-free localization is proposed in this article. The average hop distance calculated method is improved of the traditional DV-hop algorithm, and the APC algorithm is proposed to optimize the localization results of the DV-hop algorithm, which improves the localization accuracy of unknown nodes. However, many errors and questions in this work are as follows:

1.       The mathematical model of the artificial plant community mentioned in the abstract was not elaborated.

2.     Please analyze theoretically the computational complexity of the proposed algorithm compared to other methods.

3.     Please elaborate the theoretical basis that the plant community algorithm how to improve the localization accuracy of DV-hop algorithm localization results and further perform simulation.

4.     Only the node localization result of DV-hop algorithm is used as the input of APC algorithm in the algorithm flow chart, please explain the source of anchor node in APC algorithm.

5.     The plant community algorithm growth operation and environmental distinction are both expressed by the fitness function in this article, and it is recommended to introduce them differently.

6.     What are the advantages of the plant community localization algorithm over the artificial intelligence algorithm?

7.     The research of this article is range-free localization methods, but the background and related parts are mostly range-based localization methods, which are redundant and logically unclear.

8.     The examples of artificial intelligence algorithms do not clarify the subordination of various algorithms, and it is recommended to make changes. (line72-73,192-193) 

9.     Please elaborate the rationality of the population size m=20 in the APC algorithm.

10.  The target localization calculation function is not proposed in the initialization stage of APC algorithm in this article, it is suggested to explain it.

11.  Please explain the meaning of each parameter of Equation 19.

12.  Background and related works section suggests that the localization accuracy of range-free localization method is better than that of range-based localization method, which is not consistent with the previous section. (line 180-181)

13.  The authors should check the explanatory vocabulary of nouns (line 47-48, grammarline 8-9,17-18,446-447,537, clause construction (line 8-9,17-18) in this article.

14.  Figure 4 (b) and figure5(b) are inconsistent with the discussion results.

Author Response

Thank you for your review. There are too many modifications. Please refer to the attachment.

Reviewer 2 Report

The topic is interesting and timely. My major concern is regarding the interpretation of figures 5 and 6. Figure 6(a) shows the scenario of the communication range vs. computing time (200 sensor nodes, 20% anchor nodes, population size m=20). Here, the communication range r varies from 10m to 40m, and the increase will affect the computing time. As we can see from figure 6(a), the positioning computing time of all algorithms tends to increase with increasing communication range because of more connection and more complex topologies. Furthermore, it can be noticed that the proposed algorithm has less computing time, but the CNN algorithm cost too much computing time. At the same positioning accuracy, the proposed method outperforms CNN algorithm and other algorithms in terms of computing time. More explanations are needed regarding the alternatives and the rationales. Moreover, the lanugage is very poor. In some places it is very difficult to follow what the authors would like to express. I strongly suggest a professional editing service if possible. 

To improve this paper, I have some further minor comments as follows:

(1)  For introduction section, Literature review should be more detailed and comprehensive. Authors should add more recent research progress to this part and give a brief introduction of the development history on your topic.

(2) It is better for authors to clarify your objectives of your study in the Introduction section.

(3) For your proposed method, advantages compared to other methods should be clarified.

(4) Reasons why you choose this method should be added to your manuscript.

(5) In the comparative study, authors can use other methods to analyze the same problem and highlight your method's accuracy.

(6) For conclusions, more detailed results should be presented.

Author Response

(The authors gave the same response as above.)

Reviewer 3 Report

The topic is interesting, but I would suggest the following comments for improvements:

- Please improve and organize the review of related works, analyzing additional papers published in 2021 and 2022.

- In result analysis, the author must explain the results in detail, with observations and general reasons. It is beneficial that the author includes the scientific basis for each finding.

- What is the outcome's substantial impact on the research field?

- Please, explain how the proposed solution and the obtained results can prove that the research problem has been solved. 

- It is helpful that the author presents a real frank account of the proposed research method's strengths and weaknesses.

- The conclusion section seems to rush to the end. The Conclusion is really much too short, and the author will have to describe the research's impact and insights. The author necessitates rewriting the entire Conclusion section with a focus on the manuscript's implications and insights.

Author Response

(The authors gave the same response as above.)

Reviewer 4 Report

This paper discusses a novel algorithm for solving the positioning problem of wireless sensor networks inspired by the behavior of artificial plant communities. The algorithm is based on three basic operations and a fitness function and is demonstrated to achieve accurate results in limited time with limited computing resources.

This paper makes three main contributions. Firstly, it establishes an evolution model of an artificial plant community to solve the positioning problem of WSNs, using a fitness function to identify anchor nodes and position unknown nodes. Secondly, it presents an artificial plant community algorithm that simulates the growth process of a real plant community, using seeding, growing, and fruiting operations to solve the fitness function and accurately position unknown nodes. Finally, the APC algorithm is validated through experiments and compared with other AI positioning algorithms, with the results analyzed and discussed.

A new positioning algorithm, named artificial plant community (APC), is introduced in this work to enhance the accuracy of positioning in WSNs. The APC algorithm employs a weighted DV-Hop algorithm in the first phase to estimate the positions of unknown nodes, and then utilizes an evolutionary plant community to minimize positioning error. Simulation experiments are conducted to test the accuracy, iterative error, and computing time while varying network parameters.

Results indicate that the proposed APC algorithm outperforms other algorithms in terms of accuracy, error, and computation time. The APC algorithm is expected to have broad applications.

Approximately 860 words as sequences were copied from: Abdelali Hadir, Younes Regragui, Nuno M. Garcia. "Accurate Range-Free Localization Algorithms Based on PSO for Wireless Sensor Networks", IEEE Access, 2021 DOI: 10.1109/ACCESS.2021.3123360 .  This paper is not cited in the References! 

Author Response

(The authors gave the same response as above.)

Round 2

Reviewer 1 Report

None

Reviewer 2 Report

The paper can be accepted. However, I suggest the English language should be improved. 

Reviewer 3 Report

My comments have been addressed. So, the paper can be accepted.